# Evolutionary Genetics of Mycobacterium Tuberculosis and HIV-1: “The Tortoise and the Hare”

**DOI:** 10.3390/microorganisms9010147

**Published:** 2021-01-11

**Authors:** Ana Santos-Pereira, Carlos Magalhães, Pedro M. M. Araújo, Nuno S. Osório

**Affiliations:** 1Life and Health Sciences Research Institute (ICVS), School of Medicine, University of Minho, Campus Gualtar, 4710-057 Braga, Portugal; anaapspereira@gmail.com (A.S.-P.); carlosmagalhaes@med.uminho.pt (C.M.); id5827@alunos.uminho.pt (P.M.M.A.); 2ICVS/3B’s-T Government Associate Laboratory, 4710-057 Braga/Guimarães, Portugal

**Keywords:** HIV-1, *Mycobacterium**tuberculosis*, evolutionary genetics, lineage, subtype, genetic diversity, tuberculosis, AIDS

## Abstract

The already enormous burden caused by *Mycobacterium tuberculosis* and Human Immunodeficiency Virus type 1 (HIV-1) alone is aggravated by co-infection. Despite obvious differences in the rate of evolution comparing these two human pathogens, genetic diversity plays an important role in the success of both. The extreme evolutionary dynamics of HIV-1 is in the basis of a robust capacity to evade immune responses, to generate drug-resistance and to diversify the population-level reservoir of M group viral subtypes. Compared to HIV-1 and other retroviruses, *M. tuberculosis* generates minute levels of genetic diversity within the host. However, emerging whole-genome sequencing data show that the *M. tuberculosis* complex contains at least nine human-adapted phylogenetic lineages. This level of genetic diversity results in differences in *M. tuberculosis* interactions with the host immune system, virulence and drug resistance propensity. In co-infected individuals, HIV-1 and *M. tuberculosis* are likely to co-colonize host cells. However, the evolutionary impact of the interaction between the host, the slowly evolving *M. tuberculosis* bacteria and the HIV-1 viral “mutant cloud” is poorly understood. These evolutionary dynamics, at the cellular niche of monocytes/macrophages, are also discussed and proposed as a relevant future research topic in the context of single-cell sequencing.

## 1. Introduction

Tuberculosis (TB), a disease caused by *Mycobacterium tuberculosis* infection, is one of the top ten leading causes of death worldwide [1,2]. Among the estimated 1.4 million TB deaths in 2019, 208,000 individuals were co-infected with the Human Immunodeficiency Virus (HIV), representing approximately 30% of the Acquired Immunodeficiency Syndrome (AIDS) related deaths (from a total of 690,000 [3]). Indeed, TB represents the leading cause of morbidity and mortality among AIDS patients [1,3,4]. Several efforts have been made to prevent transmission and treat the viral infection, relying on antiretroviral therapy (ART), which contributed to the avoidance of 12.1 million deaths due to AIDS related illnesses between the years 2010 and 2019 [3]. Regarding *M. tuberculosis*/HIV co-infection, the combined effect of ART and TB treatment prevented 11 million deaths between 2000 and 2019 [1].

According to the World Health Organization (WHO) in 2019, the risk of developing TB was 15 to 21 times higher in people infected with HIV. Within the same host, HIV and *M. tuberculosis* present a synergetic relationship, leading to a faster deterioration of the immune system and, consequently, to reduced life expectancy [5]. In fact, studies have shown that low CD4^+^ T cell counts in HIV-infected patients relate with a higher probability of developing active TB [4,6,7,8], while individuals undergoing ART therapy are less prone to develop TB [4].

One of the biggest challenges concerning TB treatment is the existence of drug-resistant *M. tuberculosis* strains. In 2019, approximately 500,000 cases of resistance to rifampicin, the most effective first-line drug, were reported, from which 78% presented multidrug-resistance (MDR) [1]. Interestingly, transmission of drug-resistant strains is considered the main contributor to this phenomenon, rather than genetic alterations of the bacterial genome arising de novo within infected individuals [9].

Contrasting with several other human pathogens, *M. tuberculosis* generates smaller genetic diversity levels within the host and at the transmission chain levels [10,11,12]. Still, two *M. tuberculosis* strains can differ in thousands of single nucleotide polymorphisms (SPNs) and associate with different clinically relevant phenotypes [11,13,14,15,16,17]. In addition, analysis of *M. tuberculosis* genomes revealed a biased geographic distribution for the phylogenetic lineages of the pathogen, supporting a possible adaptation to specific host populations [13,18,19,20,21,22,23].

On the other hand, HIV-1 presents extremely high genetic variability at the intrahost and population levels [24,25]. HIV-1 subtypes and recombinant forms have a dissimilar global distribution [26,27], and present distinct cell tropism, viral fitness and plasma viral loads, which may play a role on the infection dynamics, disease progression and in the response to ART [28,29]. Moreover, HIV-1 genetic diversity is behind the emergence of drug-resistant viruses and underlies the biggest challenges to find an effective treatment or cure to HIV-1 infection [30].

In this review, we describe the genetic diversity and evolution of *M. tuberculosis* and HIV-1, with particular emphasis on its biomedical implication and impact in host–pathogen interactions. We then propose the evolutionary dynamics of the interaction between the host, *M. tuberculosis* and HIV-1, at the cellular niche of monocytes/macrophages, as a future research challenge to the field.

## 2. Genetic Diversity of *M. tuberculosis*

### 2.1. M. tuberculosis Lineages: Origin and Host-Pathogen Associations

The *Mycobacterium tuberculosis* complex (MTBC) is composed of obligate pathogens with no known environmental reservoir [20,21,31]. *Mycobacterium* spp. from the MTBC share high levels of DNA sequence identity, which leads to controversial taxonomy and nomenclature [32,33]. Despite this, members of the MTBC differ widely in host range and geographic distribution [14,18,19,20,31,32]. There is a clear interspecies difference in the ability to cause active disease, with animal-adapted bacteria such as *M. bovis* or *M. orygis* being isolated occasionally in humans and associated with extra-pulmonary forms of the disease [20,31,34]. On the other hand, although the human pathogen *M. tuberculosis* has been occasionally isolated in other animals [35], associated pathology in the lungs and lymph nodes of cattle was found to be very low comparatively to *M. bovis* [36].

Human-adapted MTBC bacteria are divided into at least nine distinct lineages, as determined by whole-genome phylogenetic analysis using clinical isolates from several world regions [13,37,38]. These lineages are thought to have originated several millennia ago in the Neolithic period [39], with their phylogeographic pattern subsequently defined by sociodemographic events [13,22,40,41] and biological factors [10,15,32,42,43,44]. The “ancient” lineages 1, 5, 6 and the newly proposed lineage 9 (based on five genomes isolated from East African countries [38]) were considered the first to split from the common ancestor, while the “modern” lineages (lineages 2, 3 and 4) diverged later. Lineage 1 strains are more abundant in East Africa, Southeast Asia, South Asia and Oceania; lineage 2 in Eastern Asia (e.g., China); lineage 3 in East Africa, Southwest Asia, Central and South Asia; lineage 4 in Europe, Africa and America; lineages 5 and 6 are typically restricted to West Africa and lineage 7 was only isolated in Northern Ethiopia (Figure 1 and [13,45,46]). Additionally, two *M. tuberculosis* genomes from Rwanda and Uganda highlighted the existence of a sister clade to the MTBC (i.e., with an immediate common ancestor), which was named Lineage 8 [37]. The geographic distribution of these phylogenetic lineages across the globe is not likely to be a mere consequence of founder effect. Instead, it seems to be partially driven by the interaction with specific human populations [13,20,21]. Strikingly, the phylogeographic profile of *M. tuberculosis* is maintained even in highly urbanized settings, as demonstrated for San Francisco (USA). In this metropolis, the active TB cases in immigrant communities were predominantly caused by strains belonging to lineages that were characteristically abundant in the patient’s country of origin, rather than the highly prevalent lineage 4 strains [18,47]. On a similar note, the predominant sublineages in bordering areas of Southeast Asia were found to be different, contrasting with the high level of interregional flows of people [48]. This fact suggests that some level of host–pathogen co-adaptation exists and is important to the development of active TB. *M. tuberculosis* lineages are also substantially different in terms of human host population range, which might reflect on the relative success to cause active disease. Indeed, lineages 2 and 4 cover considerably more territory than the remaining lineages (Figure 1). Moreover, lineage 4 was isolated in patients from all inhabited continents. Nevertheless, certain sublineages of lineage 4 are more geographically restricted than others [23]. In addition, robust phylogenetic dating revealed the existence of these “specialist” sublineages over many centuries [46], demonstrating that there was enough time for a more global spread in light of globalization and airborne transmission. Interestingly, “specialist” sublineages presented lower levels of sequence diversity in experimentally validated T cell epitopes [23], indicating a possible role for human adaptive immune system in the definition of host-pathogen combinations that can more likely result in active TB after infection [23].

### 2.2. Biomedical Implication of M. tuberculosis Genetic Diversity

The population structure of *M. tuberculosis* was shown to influence several biomedically relevant aspects, such as transmissibility, disease severity, drug resistance and immune response (as extensively covered in previous reviews [10,15]). Here, we describe evidence of the interaction of *M. tuberculosis* with distinct human genetic backgrounds and HIV-1.

Transmissibility of a pathogen might ultimately dictate its prevalence among host populations. The transmission potential of “modern” MTBC lineage strains has a tendency to be higher than in their “ancient” partners [15,52]. Increases in relative prevalence have been consistently reported in several world regions for Lineage 2 [15,53,54,55]. In agreement, three quarters of the transmission clusters in Ho Chi Minh City (Vietnam) involved recently introduced lineage 2 strains [54]. Importantly, lineage 2 strains progressed to active disease in younger people and in shorter periods of time, while the endemic lineage 1 led to increased chance of reactivation from latent TB and in older people [54]. Another study demonstrated the ability of the Beijing sublineage of lineage 2 (L2-Beijing) to spread to Africa, following a series of multiple and timely independent events [55]. This suggests that the introduction of L2-Beijing was not solely due to its increased mutation rate, previously found to augment drug resistance acquisition in vitro [56]. Overall, both host and bacterial factors might be involved in the colonization of different ecological niches by *M. tuberculosis* lineages. One interesting approach is to dissect this issue in light of the large disease spectrum of TB. TB manifestations can range from latency or subclinical disease [57] to increased severity [17]. To bring clarity to these topics, some authors are addressing the host and pathogen simultaneously. For instance, a significant association was reported between a nonsynonymous substitution (G254K) on the Arachidonate 5- Lipoxygenase (ALOX5) gene and increased predisposition to develop TB due to lineage 6. Accordingly, the variant G254K of ALOX5 is highly frequent in Africa when compared to other world regions [58], indicating that lineage 6 strains (abundant in West Africa) might have a selective advantage in hosts harboring the referred polymorphism. In a genome-wide association study (GWAS) with nearly 700 *M. tuberculosis* clinical isolates, a SNP near the leukocyte surface antigen CD53 gene was significantly associated with older age onset TB cases caused by lineages other than lineage 2 [59]. Interestingly, results are in agreement with the different transmission dynamics by lineage described for Ho Chi Minh City (Vietnam) [54]. On the other hand, recent work [17] could not find a single SNP explaining all the association between severe TB and *M. tuberculosis* clinical isolates inducing low levels of IL-1β secretion. Instead, several candidate mutations were found, with possible impacts at the protein level (ESX-1 secretion system) and transcription regulation (sigma factor sigA). These data show a complex interaction between the bacterium and the immune system of its host.

CD4^+^ T cells are crucial immune components to control *M. tuberculosis* [8]. This is particularly evident when viewed in the context of HIV-1 co-infection. A molecular epidemiology study performed by Lukas Fenner and colleagues [7] showed a significantly higher percentage of active TB infections due to allopatric *M. tuberculosis* strains (i.e., belonging to lineages not usually associated with the patient’s country of origin) in cases of HIV-1 co-infection. The statistical association was stronger in patients with lower CD4^+^ T cell counts and remained significant after adjusting for sex, age, contact with foreign populations and immune comorbidities. Thus, these results highlight the importance of studying the immune synapse between bacterial peptides and different human leukocyte antigen (HLA) class II proteins. However, a large majority of the experimentally validated T cell epitopes were described to be hyperconserved across the MTBC [60,61]. Nonetheless, amino acid mutations in predicted CD4^+^ T cell epitopes were identified in a comprehensive immunoinformatic screening aiming at highly diverse genes. Furthermore, amino acid variants in 10 out of 14 experimentally tested epitopes led to differential immune responses, as measured by IFN-y secretion in the blood of 82 individuals from the Gambia, latently infected with *M. tuberculosis* [61]. Other elaborated approaches have been also uncovering relevant CD4^+^ T cell epitopes, including some that are highly prevalent in certain MTBC lineages [62,63,64]. Another important feature of the immune response in TB is the balance between pathogen clearance and immunopathology [8,10]. In this context, a recent study reported the death of a patient with Merkel cell carcinoma after the use of anti-T cell inhibitory receptor PD-1 as an immunotherapeutic strategy, possibly caused by an exacerbated immune response [65]. Overall, immune findings highlight the need for an integrative approach to diagnose and treat TB, taking into account important comorbidities such as the ones caused by HIV-1.

## 3. Genetic Diversity of HIV-1

### 3.1. HIV-1 Groups and Subtypes

The world became aware of HIV-1 in the 1980s. However, by that time there was already a large dissemination of the virus. Although it was first described in the American continent, it became evident by subsequent studies that it already had a high prevalence in the central and eastern African populations [66,67]. The genetic proximity between HIV-1 and simian immunodeficiency virus (SIV), observed in African nonhuman primates (NHPs), was a fundamental piece of evidence to understand the origin of this pandemic [68,69]. Today it is widely accepted that retroviruses can be transmitted to humans from the blood or tissues of NHPs. This exposure may be a consequence of pet handling or hunting for bushmeat [70,71]. The division of HIV-1 in several groups is a direct consequence of the genetic distance observed among them, caused by independent cross-species transmission events [72]. Groups M and O were most likely the first circulating among human populations [73]. However, the closest SIV to group M is found in chimpanzees (SIVcpz, *Pan troglodytes troglodytes*) [74], while the group O counterpart was identified in gorillas (SIVgor, *Gorilla gorilla*) [75]. Group N is also closely related with SIVcpz [74]. In 2009 a new cross-species transmission event was reported. A virus closely related with SIVgor, differing from group O, was observed in a human host giving rise to the group P [76]. Later, more cases of this novel group were reported [77]. Although four different HIV-1 groups exist (M, N, O and P), only M caused a pandemic. As previously mentioned, group P was only observed in sporadic cases [77], N is largely confined to Cameroon [78] and O can be found mainly in west and south-east African countries [79]. Consequently, the term “HIV-1” is commonly used when referring to HIV-1 group M, and only when referring to other groups is the corresponding letter specified. By using molecular clock estimations, it was possible to date the early spread of HIV-1 group M and O among human populations to the 1920s [73] in the Congo River basin. Faria et al. [73] estimated the HIV-1 population growth with a two-phase exponential-logistic model, concluding that between 1920 and 1960 both group M and O had relatively slow exponential growths and were restricted to the Kinshasa area. However, around 1960, the growth rate of group M more than doubled while group O remained practically unchanged. The success of group M is most likely due to the unfortunate coincidence of several viral characteristics and human sociodemographic factors. The Vpu protein in group M seems to be more efficient regarding immune system evasion, leading to higher viral replication rates [80]. Historical records support the massive use of unsterilized injection material in the Kinshasa region, which is thought to be the cause of a hepatitis outbreak in the early 1950s [81]. Moreover, the sex commerce in the region saw several alterations in the 1960s—namely, the increased number of clients per sex worker [81]. These and other factors may have led to the observed superior exponential growth of HIV-1 group M which resulted in today’s pandemic.

The HIV-1 diversity is not restricted to group differences. Most of the modern-day cohort characterizations focus on the HIV-1 group M subtype level [82,83,84]. There are nine subtypes (A–D, F–H, J, K) some of which include sub-subtypes (A: A1, A2, A3, A4, A6 and F: F1, F2). However, some literature suggests the possibility of other severely underreported subtypes [85]. The vast genetic diversity of this pathogen is a consequence of its capacity to generate high viral loads while maintaining high mutation and replication rates [86,87,88]. The ability of the virus to rapidly accumulate changes has a tremendous effect on its success, allowing it to quickly adapt to a new host environment [72]. The generation of mutations in the HIV-1 genome is largely attributed to the viral Reverse transcriptase (RT), which has no proofreading activity. Retroviruses have two copies of the genomic RNA per virion. During the process of reverse transcription, portions of both RNA molecules can be used to generate the resulting DNA molecule. Thereafter, retroviruses have high recombination rates—among them, HIV-1 has one of the highest [89,90]. Recombination may occur between two viral RNA copies of the same HIV-1 group M subtype, and after integration in the host genome, viral particles of the same subtype will be created from the resulting DNA. However, cases of multiple infections by viruses of different subtypes may lead to the emergence of recombinant variants [91]. These are generally called HIV-1 group M recombinant forms. If only sporadic cases of a given form are observed, the nomenclature Unique Recombinant Form (URF) is used. However, if a transmission chain is originated, leading to the appearance of a high number of infections by the new recombinant form this will be denoted as a Circulating Recombinant Form (CRF) [92]. The number of identified HIV-1 group M CRFs is continuously increasing. At the time of writing, the HIV-1 sequence database has recorded more than 90 CRFs [93].

The classification of HIV-1 in groups and subtypes is extremely useful in terms of epidemic surveillance. HIV-1 subtyping tools [94,95,96,97] are user friendly and can be used to quickly infer the HIV-1 subtype from the viral sequence. The subtype information can be used as a proxy of diversity. For instance, the increase in a rare subtype prevalence among new infections may be the indication of new patterns of transmission or behavior in a given population. Moreover, observing a large number of recent infections by the same subtype may be an indicator of a vast transmission chain that must be identified and tackled [83,98]. The nomenclature proposal made by Robertson et al. [92] was a core document to normalize and created a reference classification for HIV-1.

The global distribution of HIV-1 subtypes is uneven (Figure 1). It is estimated that subtype C accounts for almost 50% of the HIV-1 infections worldwide, being more prevalent in the African continent. The most well studied subtype is B, a result of its high prevalence in the Americas, Europe and Australia [82,99]. Since it is the geographic origin of HIV-1, the countries in the central east region of Africa have the highest subtype diversity.

### 3.2. HIV-1 Viral “Quasispecies”—HIV-1 Intra Host Diversity

Most HIV-1 hosts were exposed to a single transmission event. Nevertheless, in what is thought to be a relatively rare occurrence the same host can be exposed to multiple HIV-1 transmission events [91]. The term dual infection is sometimes applied referring to multiple infections, which should be done thoughtfully since it assumes only two events. However, dual infection is a practical term when referring to the “previous” and the “following” infections. Therefore, dual infection can be divided into co-infection or superinfection. Co-infection refers to two infection events occurring within a brief period. On the other hand, a superinfection refers to a subsequent infection after the establishment of a complete immune response to the first event [91,100]. The different infection events can harbor similar or very distinct HIV-1 genomes. The presence in the same host of viruses from different HIV-1 group M subtypes raises the potential of recombination among them generating recombinant forms [101]. Even recombination among viruses from the same subtype can aid the virus to successfully adapt to the host. Different viral particles, in the same host, can harbor beneficial genetic characteristics, which when combined (due to recombination) can lead to the emergence of a virus with greater fitness. The process of selecting successful viral variants occurs in a trial and error manner and is only possible due to the large viral load generated by HIV-1 infection [101,102]. This within host variability is sometimes referred to as “quasispecies”. This term attributed to Eigen [103,104] describes the pool of different, yet similar, genomes accumulating genetic variations, competing, and being selected within host [105]. This “cloud” of viral genomes allows the viral population to escape, or at least partially escape, host immune and antiretroviral treatment (ART) pressures.

### 3.3. Biomedical Implication of HIV-1 Genetic Diversity

HIV-1 infections by groups N, O and P have clinical manifestations distinguishable from the pandemic group M. However, their occurrence is rather rare [72,74,75,77,78,79]. Several studies focused on enlightening the impact of the M group subtype diversity in clinical correlates. It was demonstrated that subtype A leads to a slower decline of CD4^+^ T cell counts and consequently slower disease progression when compared, in several cohorts, to subtypes C and D [106,107,108]. This seems to be independent of viral load [108,109]. Additional comparisons between these three subtypes suggest a preference for in utero transmission of subtype C [110,111]. The genetic differences observed among subtypes can sometimes correspond to sites targeted by ART. Of note, ART was largely developed focusing on subtype B, due to its geographic distribution [112]. Therefore, differences in the prevalence of certain polymorphisms can lead to a differential response to ART. This was observed for all the viral proteins targeted by treatment so far [113,114,115]. Although subtype differences may affect treatment susceptibility, their impact is limited, and ART is expected to be efficient across subtypes [116]. CD8^+^ T cells have a crucial importance in antiviral immunity [117], and consequently the viral regions encoding epitopes recognized by these cells are known to accumulate escape mutations [118,119]. Similarly to what was previously described for the ART, the subtype diversity can impact the immune response capacity [120]. As previously reviewed in more detail by Carlson et al., having a higher (several mutations) or lower (few mutations) barrier to the appearance of the escape variants when exposed to a given set of immune pressures can have an impact on the infection progression of the different HIV-1 subtypes [119,121].

Viral cellular reservoirs represent one of the biggest challenges towards the cure of HIV-1 infection, as the virus is able to maintain residual replication levels even during ART treatment, chronically compromising the host immune system [122,123,124]. HIV-1 genetic diversity also seems to play a role in viral preferential tropism. Differences in cellular reservoirs preference in the lymph nodes have been observed by Bhoopat and colleagues [125,126] between HIV-1 subtypes B and CRF01_AE (previously known as E subtype). A more recent study also demonstrated that subtype B established larger reservoirs when compared with the subtypes CRF01_AE and G, with larger reservoirs related to worse prognosis [127].

Overall, the genetic diversity of HIV-1, represented in the form of groups, subtypes, and recombinants, is the largest obstacle for the creation of an effective and comprehensive strategy to prevent or cure AIDS [119,128,129].

## 4. Conclusions and Future Perspectives

Infectious diseases were traditionally investigated in an isolated or pairwise fashion considering a single pathogen and the host. However, humans are in constant interaction with the microbiome and are often infected by multiple pathogens. Moreover, host–pathogen interactions consist of a complex network of protein–protein interactions that influences the evolution of the interrelating entities at each given time. In the cases of *M. tuberculosis* and HIV-1, both gain reciprocal advantages in co-infecting the human host. Nevertheless, it remains elusive how and to what extent the evolving community of viruses and bacteria within an infected host can communally shape genetic diversity and evolutionary patterns.

Since macrophages are the primary intracellular niche for *M. tuberculosis* and are also permissive to HIV-1 infection, it is likely that the two pathogens can modulate/compromise these cells, facilitating co-infection and contributing to accelerate the progression of both diseases.

The most important co-receptors for HIV-1 binding and cell entry are CCR5 and CXCR4. Virions can display affinity for both receptors or increased affinity to either CCR5 or CXCR4. For reasons that are not fully understood and regardless of the infection route, CCR5-tropic viruses are the most transmitted [130]. Thus, CCR5 receptors are considered to play a central role in the initial stages of HIV-1 infection. In addition to memory CD4^+^ T cells, CD4 and CCR5 are also expressed in monocytes and macrophages [131]. However, efficient replication of HIV-1 within infected macrophages is limited by the restriction factor SAMHD1 [132]. The activity of SAMHD1 is downregulated when macrophages are in a G1-like state allowing HIV-1 to bypass SAMHD1 restriction [133]. Interestingly, it has been shown that *M. tuberculosis* infection inhibits the macrophage G1/S transition arresting the macrophages in the G0-G1 phase [134]. Thus, it is tempting to speculate that *M. tuberculosis*-infected macrophages become more permissive to allow HIV-1 replication. Possible consequences of higher HIV-1 viral loads in the infected macrophages are an increase in the intrahost viral genetic diversity and a more rapid conversion from an initial population of CCR5-tropic viruses to a mixed population of CCR5- and CXCR4-tropic viruseses, expanding the possibilities for the virus to target other host cells.

On the other hand, the increased severity of TB in HIV-1/*M. tuberculosis* co-infected individuals was shown to be dependent on CD4^+^ T cell counts [135]. Since CD4^+^ T cells are the primary target cells of HIV-1 and are also central for the control of *M. tuberculosis* infection [136,137], it is very likely that the aggravation of TB due to HIV-1 infection is caused by a decrease in the host protective responses against *M. tuberculosis*. Furthermore, the large increase in the risk of developing active TB in HIV-infected individuals is present even at early stages of infection with normal CD4^+^ T cells counts [138,139]. This suggests that other factors might be involved but it is also relevant to consider that during the progressive depletion of CD4^+^ T cells, less frequent clones might become absent even at levels of total CD4^+^ T cells that are considered normal. Fenner and colleagues [7] showed that in HIV-1/*M. tuberculosis* co-infected individuals, there is a significant increase in cases of active TB due to *M. tuberculosis* strains belonging to lineages that do not usually occur in that particular geographic region [7]. *M. tuberculosis* lineages show high levels of conservation in T cell epitopes. Nonetheless, outlier epitopes to the general rule of hyperconservation have been described [61,140]. It was shown that because of this there is a higher degree of host-pathogen variability in T cell responses than previously expected [61]. Thus, it is conceivable that during the HIV-1 inflicted depletion of CD4^+^ T cells, the capacity of the host to control specific lineages of *M. tuberculosis* and prevent the transition from latent to active TB is compromised.

Overall, the interaction between HIV-1, *M. tuberculosis* and their obligate human hosts is determined by a complex network of host, pathogen and environmental factors. We propose that in co-infected hosts, HIV-1 and *M. tuberculosis* mutually influence their genetic evolutionary dynamics. This is likely to occur in potentially overlapping intracellular niches, such as monocytes/macrophages, and via CD4^+^ T cells. Thus, we highlight single-cell RNA sequencing (scRNA-seq), which has been highlighted as a valuable tool to study both viral and bacterial interactions with the host [141,142,143], as a technique to be used in future studies since it could provide crucial insights into the synergy of HIV-1 and *M. tuberculosis* within the same cellular niches.

Future integrated strategies to further understand this crosstalk between HIV-1 and *M. tuberculosis* can help in disease control and in the clinical and epidemiological stratification of high-risk populations.

## Figures and Tables

**Figure 1 microorganisms-09-00147-f001:**
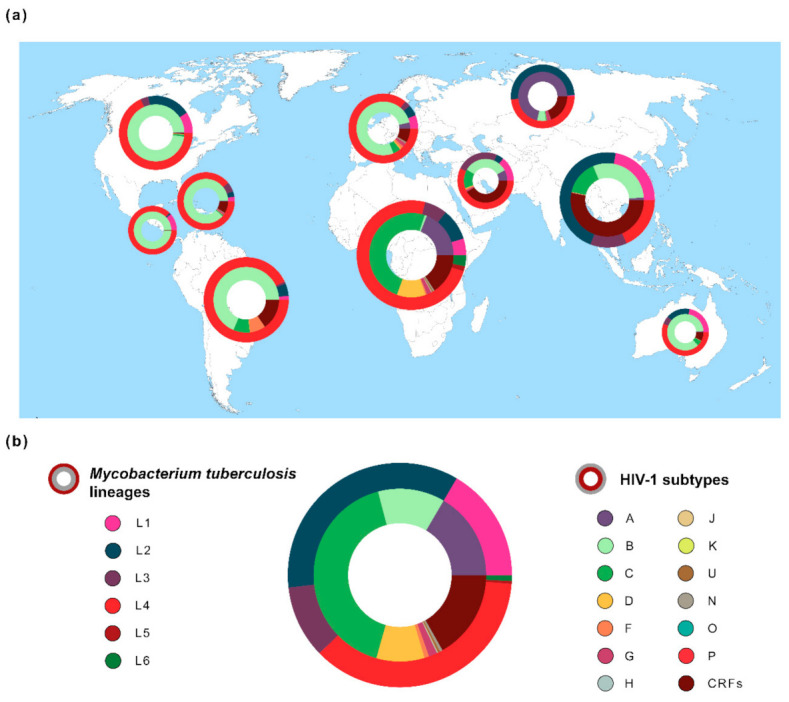
Global distribution of the genetic diversity of *Mycobacterium tuberculosis* and Human Immunodeficiency Virus typ1 1 (HIV-1). (**a**) Frequencies of *M. tuberculosis* lineages (outer ring/circle) and HIV-1 subtypes (inner ring/circle) across 10 geographic regions. Pie chart size is related with both diseases’ prevalence in each region [1,3]. Geographic regions are according to Los Alamos HIV database (http://www.hiv.lanl.gov/) [25]. *M. tuberculosis* spoligotype clades (*n* = 64,555 sequences) were extracted from the SITVIT2 database [49] and converted to *Mycobacterium tuberculosis* complex (MTBC) lineages 1–6 [45,50,51]. HIV-1 data (*n* = 815,431 sequences) were obtained from Los Alamos HIV database [25]. (**b**) Global frequency of *M. tuberculosis* lineages and HIV-1 subtypes. The estimated number of people living with TB or HIV-1 was obtained from the World Health Organization (WHO) [1] or from the Joint United Nations Programme on HIV/AIDS (UNAIDS) [3].

## Data Availability

Not applicable.

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
