# Peer review of "Evolutionary Genetics of Mycobacterium Tuberculosis and HIV-1: “The Tortoise and the Hare”"

_microorganisms, 2021, doi:10.3390/microorganisms9010147_

Round 1
Reviewer 1 Report
I strongly recommend the publication of this manuscript in your journal as it is.
In my opinion it represents a great piece of work which provide significant contribution about the co-infection by M. tuberculosis and HIV-1.
I have no suggestion for authors
Author Response
We thank for the time dedicated in reviewing the manuscript and for the very positive feedback presented on our work.
Reviewer 2 Report
This is a well written review of the genetic evolution of M. tuberculosis and HIV-1. It describes the clades and subtypes and provides some ideas on the co-evolution of these human pathogens. It contains a very nice and comprehensive illustration on a world map of the different lineages of Mtb and the subtypes of HIV-1.
The weaker part is where the impact of ART is very briefly discussed; I am not sure this provides an in depth discussion, but I also realize that it would require adding a large portion of text and novel references to be added, in order to provide a comprehensive review of drug treatment on the different subtypes; I am also not fully confident that there is much literature in this field.
Detailed suggestions for further improvement:
Line 89 biologic factors: perhaps better, biological factors.
Line 101 last word: emigrant – consider changing into immigrant
Lines 163-4: . . IFN-gamma secretion in the blood of individuals from the Gambia, latently infected with M. tuberculosis . . (patients are not latent; and LTBI are not patients, they are in fact healthy individuals . . )
Line 215 During the reverse transcription, portions . . (add ‘comma’ for clarity)
Line 217: please rephrase; . . have high recombination rates, among them . . change ‘comma’ into semicolon: . . have high recombination rates; and HIV-1 has .
Line 219 genome, the resulting (add ‘comma’)
Line 222 . . are observed, the nomenclature (add comma)
Line 228 consider rephrasing: . . are easy to use and generating results rapidly, .
Line 256 . . is sometime referred to as (add ‘to’)
Line 263 focused on (not in)
Line 272 treatment susceptibility, their impact (add ‘comma’ for clarity)
Line 273 not sure I get it; consider rephrasing; ‘The host immune response can also see its effectiveness vary . . ‘ Does the host immunity have eyes to see? Do the authors mean to say, that host immunity is differentially affected by differences in viral genetics? Are they sure this is true? In my simple mind, all depends on viral suppression, and once ART is effective enough (compounded by appropriate cART selection and adherence to this optimal cART), host immunity will restore, irrespective of the viral genome . . But I will be happy to be convinced of anything else, if the authors provide the required evidence from the literature.
Line 275: . . recognized by them . . change into: . . by these CD8+ T cells
Line 277-9 here I get lost; if the authors intend to review the entire field of genetic mutations resulting in loss of cART efficacy, this would need them to embark on a huge field of treatment challenges guided by genetic analysis of genetic targets. As it stands, I feel that this sentence is not appropriate; the authors might provide detailed examples of genetic mutations resulting in loss of efficacy, in turn resulting in low-level or high-level viral replication, and subsequent drop in CD4+ count; my guess is that they might then refer to reviews that address this problem in detail.
Lines 289-3 also address this huge problem, in a fashion that does not convince the reader; as it stands, it is rather superficially addressing the issue of failure in viral suppression as a result of mutant subspecies. ‘ as preferential tropisms seem to differ. . . without a reference, it makes me feel uneasy. My suggestion is: rephrase or delete . .
Line 296 . . both organisms . .HIV-1 is NOT an organism (virions are not living organisms) so consider change into ‘pathogen’
Author Response
We would like to thank the reviewer for the time dedicated to carefully reading the manuscript and for the thoughtful insights. We have fully addressed the detailed suggestions: "Line 89 biologic factors: perhaps better, biological factors." The alteration was performed. "Line 101 last word: emigrant – consider changing into immigrant" The term “emigrant” was replaced by “immigrant”. "Lines 163-4: . . IFN-gamma secretion in the blood of individuals from the Gambia, latently infected with M. tuberculosis . . (patients are not latent; and LTBI are not patients, they are in fact healthy individuals . . )" We agree with the reviewer and changed the sentence accordingly. "Line 215 During the reverse transcription, portions . . (add ‘comma’ for clarity)" The alteration was performed. "Line 217: please rephrase; . . have high recombination rates, among them . . change ‘comma’ into semicolon: . . have high recombination rates; and HIV-1 has . " We agree with the reviewer and changed the sentence accordingly. "Line 219 genome, the resulting (add ‘comma’) " The alteration was performed. "Line 222 . . are observed, the nomenclature (add comma) " The alteration was performed. "Line 228 consider rephrasing: . . are easy to use and generating results rapidly, . " We agree with the reviewer and the sentence was rephased. "Line 256 . . is sometime referred to as (add ‘to’) " The alteration was performed. "Line 263 focused on (not in) " The alteration was performed. "Line 272 treatment susceptibility, their impact (add ‘comma’ for clarity) " The alteration was performed. "Line 273 not sure I get it; consider rephrasing; ‘The host immune response can also see its effectiveness vary . . ‘ Does the host immunity have eyes to see? Do the authors mean to say, that host immunity is differentially affected by differences in viral genetics? Are they sure this is true? In my simple mind, all depends on viral suppression, and once ART is effective enough (compounded by appropriate cART selection and adherence to this optimal cART), host immunity will restore, irrespective of the viral genome . . But I will be happy to be convinced of anything else, if the authors provide the required evidence from the literature. " This sentence was removed. "Line 275: . . recognized by them . . change into: . . by these CD8+ T cells " The alteration was performed. "Line 277-9 here I get lost; if the authors intend to review the entire field of genetic mutations resulting in loss of cART efficacy, this would need them to embark on a huge field of treatment challenges guided by genetic analysis of genetic targets. As it stands, I feel that this sentence is not appropriate; the authors might provide detailed examples of genetic mutations resulting in loss of efficacy, in turn resulting in low-level or high-level viral replication, and subsequent drop in CD4+ count; my guess is that they might then refer to reviews that address this problem in detail. " We agree with the reviewer and referred to a review addressing this topic in detail. "Lines 289-3 also address this huge problem, in a fashion that does not convince the reader; as it stands, it is rather superficially addressing the issue of failure in viral suppression as a result of mutant subspecies. ‘ as preferential tropisms seem to differ. . . without a reference, it makes me feel uneasy. My suggestion is: rephrase or delete . ." These sentences were rephrased following the suggestion from the reviewer. "Line 296 . . both organisms . .HIV-1 is NOT an organism (virions are not living organisms) so consider change into ‘pathogen’" The term “organisms” was replaced by “entities” to refer more accurately to HIV-1, Mycobacterium tuberculosis and the host.